# Assessment of patient perceptions of counselling on oral antineoplastic agents by a dedicated cancer services pharmacist in an outpatient cancer clinic

Lorna McNabb[1], Eva Metrot[1], Micaela Ferrington[1], Bruce Sunderland[1], Richard Parsons[1], Tandy-Sue Copeland[2], Siobhan Corscadden[2], Selina Tong[2], Petra Czarniak[1] *

1 Curtin Medical School, Faculty of Health Sciences, Curtin University, Bentley, Western Australia, Australia,
2 Pharmacy Department, Fiona Stanley Hospital, Murdoch, Western Australia, Australia

* P.Czarniak@curtin.edu.au

**Data Availability Statement:** The dataset can be found at https://doi.org/10.25917/2ZAB-V08 It is

## Abstract

### Background

Oral antineoplastic agents have caused a paradigm shift in cancer treatment, however, they produce many unique challenges. Although oral antineoplastics can have complex administration regimes, low adherence rates and high possibilities of drug-drug interactions, they are administered unsupervised at home. Cancer services pharmacists have the required skillsets to improve patient outcomes associated with oral antineoplastic treatment by increasing patient health literacy, improving concordance and optimising administration protocols.

### Aim

To evaluate patients' perceptions, experiences and overall satisfaction with dedicated clinical pharmacist consultations in patients treated with oral antineoplastic agents at a major public hospital.

### Method

In this retrospective cross-sectional study at a quaternary hospital in Western Australia, data were collected by a paper questionnaire (mailed in March 2022) to a random sample of 191 patients initiated on oral antineoplastic drugs between January 2021 and February 2022. Demographics, prescribed antineoplastic drug/s, cancer type data were collected including using 5-point Likert scale questions assess patients' overall satisfaction with the clinical pharmacist consultations.

### Results

The questionnaire response rate was 27.7% (52/188) (mean age 63.2 years; 57.5% female). Most patients (42/52; 80.8%) were satisfied with pharmacist consultations, trusted the pharmacist's advice (45/52; 86.5%), considered that the pharmacist improved their

labelled 'Cancer Services Pharmacist Questionnaire Data'.

**Funding:** The author(s) received no specific funding for this work.

**Competing interests:** The authors have declared that no competing interests exist.

understanding of how to manage side effects (43/52; 82.7%) and they provided an important service in outpatient care (45/52; 86.5%).

## Conclusion

Overall, patients reported positive perceptions, experiences, and satisfaction with the cancer services pharmacist counselling services during their oral antineoplastic treatment.

## Introduction

The recent explosion in the development and availability of oral antineoplastic agents (OAAs) has caused an overall paradigm shift in cancer treatment [1, 2]. Oncologists are increasingly prioritising patient preference and quality of life by choosing treatments that maximise convenience and flexibility, as well as clinical outcomes [3]. Previous studies have shown that patients display a preference for oral home-based over intravenous (IV) treatment, when efficacy is not compromised. Oral treatment allows greater convenience, with at home administration and preferable toxicity profiles [4, 5]. Additionally, OAAs can reduce the cost burden of cancer on the health care system, as patients do not require an inpatient or day patient stay [6]. This reduces the need for resources such as nurses, doctors and other health professionals [6]. Importantly, OAAs have played a positive role throughout the COVID-19 pandemic as patients undergoing cancer treatment are immunocompromised, at a higher risk of morbidity and mortality associated with the virus [7]. The flexibility of at home treatment with OAAs reduced the risk of being exposed to COVID-19 [7]. It also alleviated patient's concerns and anxieties surrounding their health [7].

While OAAs offer advantages over IV chemotherapy, they also have their own unique challenges. In the absence of supervised administration and coordination of their treatment by a doctor or nurse, patients are at higher risk of nonadherence and medication errors. OAAs dosing regimens may be complex and can contribute to the risk of nonadherence due to factors such as confusion, or a lack of understanding of their treatment regimen [6]. To maximise treatment outcomes and patient experiences, health professionals must recognise and appreciate the potential problems that can arise with the use of OAAs [6]. These niche barriers to optimising OAA therapy highlight the integral role of cancer services pharmacists (CSPs) within an outpatient cancer clinic.

CSPs have comprehensive drug knowledge and hence are in a prime position to counsel patients on the issues occurring with OAA therapy [8]. They have the required skillsets to individually educate patients on potential drug-drug interactions, how to correctly administer their OAAs, side effects and management if they occur [9, 10]. Several international studies have shown that adding specialist oncology pharmacists to cancer centres within hospital settings produces positive outcomes, since they have the expertise to provide optimal medication management [8, 11–17]. CSP interventions have identified many medication errors within the complexity of patients' OAA treatment [18]. For this reason, follow ups with a CSP may be vital in reducing these errors, enhancing patient adherence and safety and optimising therapeutic outcomes [18]. Additionally, CSPs can educate on the correct disposal and handling of OAAs, many of which are cytotoxic [19]. Patients have reported increased satisfaction and confidence following pharmacist-led education sessions [10]. This further highlights the importance of CSP in OAA therapy, as a positive patient experience is one of the key measures of successful OAA therapy.

Patient satisfaction is an emerging area of research driving improvements in health care [20]. The current literature suggests patients have reported increased satisfaction and confidence following pharmacist-led education sessions [10, 21]. Evaluating patient perceptions has been shown to be an effective method of assessing the role of the CSP and the quality of their care [20]. Although this method is more subjective than measuring tangible patient outcomes, it is representative of the reality of care provided by the CSP [20]. For this method to be effective, variables important to both the patient and the health care system must therefore be identified [20].

The professional practice standards for clinical oncology pharmacy services, as prescribed by the Society of Hospital Pharmacists of Australia (SHPA), require these services to be provided to both inpatient and outpatient oncology settings [22]. SHPA promotes pharmacist-led patient counselling and education sessions for patients undergoing OAA treatment, as part of the medical and nursing team. In response to the Cancer Plan 2023–2033 Consultation [23], SHPA commenced a two-year training program for hospital pharmacists to further increase their skill set in cancer services [23]. This program aims to increase the workforce of CSPs and allow for specialisation within the field [23]. Although global research into the impact of a CSP on outpatient OAA treatment is available and evolving, there are limited Australian based studies. A previous study conducted in Australia in 2011 demonstrated the benefit of a specialist cancer pharmacist who joined a lung cancer clinic team for six months [24]. Two recent studies in Western Australia (WA) evaluated the impact of specialist pharmacist consultations in a comprehensive Cancer Centre across a range of cancer types [25, 26]. One of the studies, which also involved an online staff survey to evaluate perceptions of health service staff about consultations provided by CSPs, reported that 97% of staff strongly agreed/agreed that CSP consultations improved patients' understanding of and confidence in managing their medications [25]. Despite improved patient outcomes reported in both studies, neither study involved a dedicated CSP. The role of a dedicated CSP is a novel role in Australia and is presently poorly researched and acknowledged.

In 2017, clinical pharmacists at a Comprehensive Cancer Centre at a leading quaternary hospital in WA began providing counselling services to cancer patients on OAAs [25]. A previous study investigated patient perceptions, experiences and satisfaction with clinical pharmacist (not dedicated CSP) consultations in patients treated with OAA at the same hospital [26]. In January 2021, a full time CSP was employed. This appointment followed recommendations made by the Australian Commission on Safety and Quality in Health Care, Clinical Oncology Society of Australia and the SHPA guidelines for OAA prescriptions to be clinically verified by a CSP prior to dispensing. The pharmacist clinic at the hospital is an outpatient pharmacist led counselling clinic run within the cancer centre. This service is available to all patients receiving OAA treatment through the hospital, patients requiring supportive care, complementary and alternative medicine reviews and those working to quit smoking. During the initial meeting, the pharmacist reviews all medications taken by the patient for drug-drug/ food-drug or disease-drug interactions, discusses potential OAA side effects (from life threatening issues, such as febrile neutropenia, to adverse effects that may affect quality of life, such as mild nausea or diarrhoea, and how best to deal with them), adherence strategies, and monitors tolerability and toxicity in clinical and laboratory tests.

The aim of this study was to evaluate patients' perceptions, experiences, and overall satisfaction with the dedicated clinical pharmacist consultations for patients treated with OAAs at the Cancer Centre.

## Methods

The WA Health Governance, Evidence, Knowledge and Outcome system (GEKO) reviewed and approved this study as a Quality Improvement Activity (study number QA33596).

Reciprocal ethics approval was also provided by the Curtin University Human Research Ethics Committee (study number HRE2021-0026 approved 27[th] January 2022.

This study was a retrospective cross-sectional study of patient perceptions of a dedicated CSP's counselling sessions at a 783-bed metropolitan quaternary tertiary hospital in WA. The questionnaire instrument consisted of four demographic questions regarding age, gender, level of education and English as first language and 16 qualitative questions addressing the benefits of the interaction between the patient and the CSP, self-perceived confidence and knowledge gained, the necessity of the role of the clinical pharmacist and personal adherence and support. These were assessed using a 5-item Likert scale anchored by 'strongly agree' and 'strongly disagree'. The questionnaire was based on a validated questionnaire previously developed and utilised by the authors PC and BS, within a similar study before the service was formally established. [26].

The hospital provided a dataset of all patients reported to have an interaction with a CSP between the 1[st] January 2021 and 28[th] of February 2022. The dataset included 1528 records and, after removing duplicates, 512 records remained. A randomiser (www.randomizer.org) was used to generate a random sample of 300 patients. Deceased patients (n = 58) were removed and a questionnaire, a patient information sheet and a reply-paid envelope was sent via Australia Post to the remaining patients. Patients who consulted with the CSP between 1[st] January 2021 and 28[th] of February 2022, were taking at least one OAA and were living, were considered eligible for this study. Patients were excluded if they were receiving exclusive IV chemotherapy or sub-cutaneous injections or were under the age of 18 years.

Data on all eligible patients was collected using the electronic medical records database at the hospital. Demographic information obtained included date of birth, gender and postcode. Other information collected included type of cancer, date of the initial consultation with the CSP, what OAAs the patient was prescribed, who referred the patient to the CSP services (doctor, nurse, pharmacist, other), reason for their initial consult, whether their consult was via telephone or in person, if there was a follow up consultation with the CSP, if a medication reconciliation was performed and whether a pharmacist intervention occurred. 'Medication reconciliation' was defined as documenting all prescription and non-prescription medications the patient was taking at the time of consultation, to allow for drug-drug/ food-drug or disease-drug interaction checking [26]. A 'pharmacist intervention' involved the CSP making a recommendation of change to the current OAA regimen, such as a dose change, drug change due to interactions and/or identification of an adverse effect. This information was transferred to a password protected Excel spreadsheet. Each patient was then assigned a unique number and the data was de-identified.

The paper-based questionnaires were mailed from the hospital on 18[th] March 2022 with a requested return date of 31[st] March 2022. Responding to the questionnaire was accepted as consent. Each questionnaire had a unique number assigned to it which was able to be linked to the patient's Unit Medical Record Number (UMRN) specific to the Western Australian Department of Health, so that returned responses could be linked to the patient's demographic data. De-identified data were entered into an Excel spreadsheet and exported to IBM SPSS. Returned responses to each question and any additional comments and/or feedback were transferred into the password protected Excel spreadsheet. The questionnaires were then securely stored at the hospital in a locked cabinet.

The data were analysed using SPSS software. Categorical data (gender, post code, who referred the patient to the CSP, cancer type, telephone or in person consultation, follow up, intervention, reconciliation) between questionnaire respondents and non-respondents were compared. Chi square and Pearson Correlation tests were used to analyse age and level of education against Likert scale responses. Fisher's exact test was used for all remaining categorical

data due to small sample sizes. For categorical analysis of age, age groups of <29, 30–49, 50–69 and >70 years of age were used. For analysis purposes the Likert scale responses were grouped as 'yes' for strongly agree and agree and 'no' for neutral, disagree, strongly disagree and no response. An independent sample $t$-test was used to compare age between respondents and non-respondents. A $p$ value of < 0.05 was considered statistically significant.

## Results

Of the 242 original patient sample, 54 patients were excluded due to not meeting the inclusion criteria. Reasons for exclusion included not taking an OAA (n = 42), no record of a consult with the CSP (n = 5), duplicate patients (n = 4), under the age of 18 (n = 1) or their consult was outside the investigation period (n = 2). From the eligible 188 patients, there was a response rate of 27.7% (n = 52).

### Patient characteristics

The mean age of the patient sample was 63.2 years, with a range of 29–92 years and a standard deviation of 13.9, with 80 (42.6%) males and 108 (57.4%) females (Table 1). There were no statistically significant differences in the characteristics evaluated between respondents and non-respondents. The most common cancer type in the patient sample was breast (28.2%), followed by leukaemia (18.6%), multiple myeloma (11.2%) and colorectal (10.1%). The remaining cancer types were grouped as 'other' (31.9%) for analysis purposes. Clinical data including cancer type, the referring clinician to the CSP (doctor, nurse, pharmacist or other) and whether the initial consult was conducted via telephone or in person were analysed between responders and non-responders. There were no statistically significant differences in these variables between respondents and non-respondents (Table 1).

Follow-up consultations, pharmacist interventions and medication reconciliations were similar for responders and non-responders (Table 1). Of those who responded, 65.4% (n = 34) had a follow up consultation, 71.2% (n = 37) had a pharmacist intervention and 80.8% (n = 42) had a medication reconciliation completed. Of those who did not respond, 55.1% (n = 75) had a follow up consultation, 72.1% (n = 98) had a pharmacist intervention and 75.0% (n = 102) had a medication reconciliation.

### Questionnaire data

The Likert scale questionnaire and responses are shown in Fig 1. There was at least 80% agreement with 12 of these questions, with lower agreement for questions 6–9. These four questions related to understanding what medical interventions would be appropriate following side-effects, support services available, confidence in visiting the pharmacist, and accessibility of the pharmacist. Eighty two percent of respondents were satisfied overall with the services that the pharmacist provided.

When examining potential associations between responder characteristics (level of education, English as a first language, gender, and postcode) or what type of clinician referred the patient to the CSP services and overall satisfaction with the pharmacist and the 4 questions showing lower agreement (above), the only questions showing significant associations were the pharmacist follow-up (for questions 6, 7 and 16).

There were no patients aged 29 or younger, 5 patients aged 30–49 years, 28 patients aged 50–69 years and 22 patients over the age of 70 years. A statistically significant difference was found between the three age categories for question 8 (I felt confident visiting my clinical pharmacist for counselling regarding my medication, p = 0.021), question 10 (overall, I trusted my clinical pharmacist's advice, p = 0.004) and question 16 (overall I am satisfied with the services

**Table 1. Responder and non-responder data (obtained from electronic patient medical record database).**

| | n (%) | Returned Questionnaire | | p-value |
|---|---|---|---|---|
| | | Yes n(%) | No n(%) | |
| **Gender** | | | | |
| Male | 80 (42.6) | 24 (46.2) | 56 (41.2) | 0.537 |
| Female | 108 (57.4) | 28 (53.9) | 80 (58.8) | |
| **Postcode** | | | | |
| Perth | 148 (78.7) | 39 (75.0) | 109 (80.2) | 0.441 |
| Other | 40 (21.3) | 13 (25.0) | 27 (19.9) | |
| **Age (years)** | | | | |
| 18–29 | 2 (1.1) | 0 (0.0) | 2 (1.5) | |
| 30–49 | 29 (15.4) | 5 (9.6) | 24 (17.7) | 0.417 |
| 50–69 | 84 (44.7) | 26 (50.0) | 58 (42.7) | |
| 70+ | 73 (38.8) | 21 (40.4) | 52 (38.2) | |
| **Referred by** | | | | |
| Doctor | 122 (64.9) | 36 (69.2) | 86 (63.2) | |
| Nurse | 32 (17.0) | 7 (13.5) | 25 (18.4) | 0.773 |
| Pharmacist | 28 (14.9) | 8 (15.4) | 20 (14.7) | |
| Other | 6 (3.2) | 1 (1.9) | 5 (3.7) | |
| **Cancer type** | | | | |
| Breast | 53 (28.2) | 13 (25.0) | 40 (29.4) | |
| Leukaemia | 35 (18.6) | 11 (21.2) | 24 (17.7) | 0.821 |
| Myeloma | 21 (11.2) | 6 (11.5) | 15 (11.0) | |
| Rectal | 190 (10.1) | 7 (13.5) | 12 (8.8) | |
| Other | 60 (31.9) | 15 (28.9) | 45 (33.1) | |
| **Initial Consult** | | | | |
| Phone | 106 (56.4) | 25 (48.1) | 81 (59.6) | 0.156 |
| In person | 82 (43.6) | 27 (51.9) | 55 (40.4) | |
| **Follow-up** | | | | . |
| Yes | 109 (58.0) | 34 (65.4) | 75 (55.1) | 0.203 |
| No | 79 (42.0) | 18 (34.6) | 61 (44.9) | |
| **Intervention** | | | | |
| Yes | 135 (71.8) | 37 (71.2) | 98 (72.1) | 0.902 |
| No | 53 (28.2) | 15 (28.9) | 38 (27.9) | |
| **Medication Reconciliation** | | | | |
| Yes | 144 (76.6) | 42 (80.8) | 102 (75.0) | 0.403 |
| No | 44 (23.4) | 10 (19.2) | 34 (25.0) | |

my clinical pharmacist offered, p = 0.031). This is shown in Fig 1. Age had no statistically significant impact on the remaining questionnaire responses.

As there was strong agreement with the questions asked, a question of importance related to the statement 'my clinical pharmacist explained to me what the likely course of action by health professionals would be if I experienced side effects', which showed that pharmacist follow-up significantly improved patients' knowledge (p = 0.0295). Advantages of pharmacist follow-up were also observed with respect to support service ie 'my clinical pharmacist helped improve my understanding of additional supportive services that could help me' (p = 0.0118). However, some improvement is necessary to increase patients' confidence to visit their clinical pharmacist for counselling with regard to their medication and accessibility of the clinical pharmacist (p = 0.5158 and 0.5103 respectively) (Table 2).

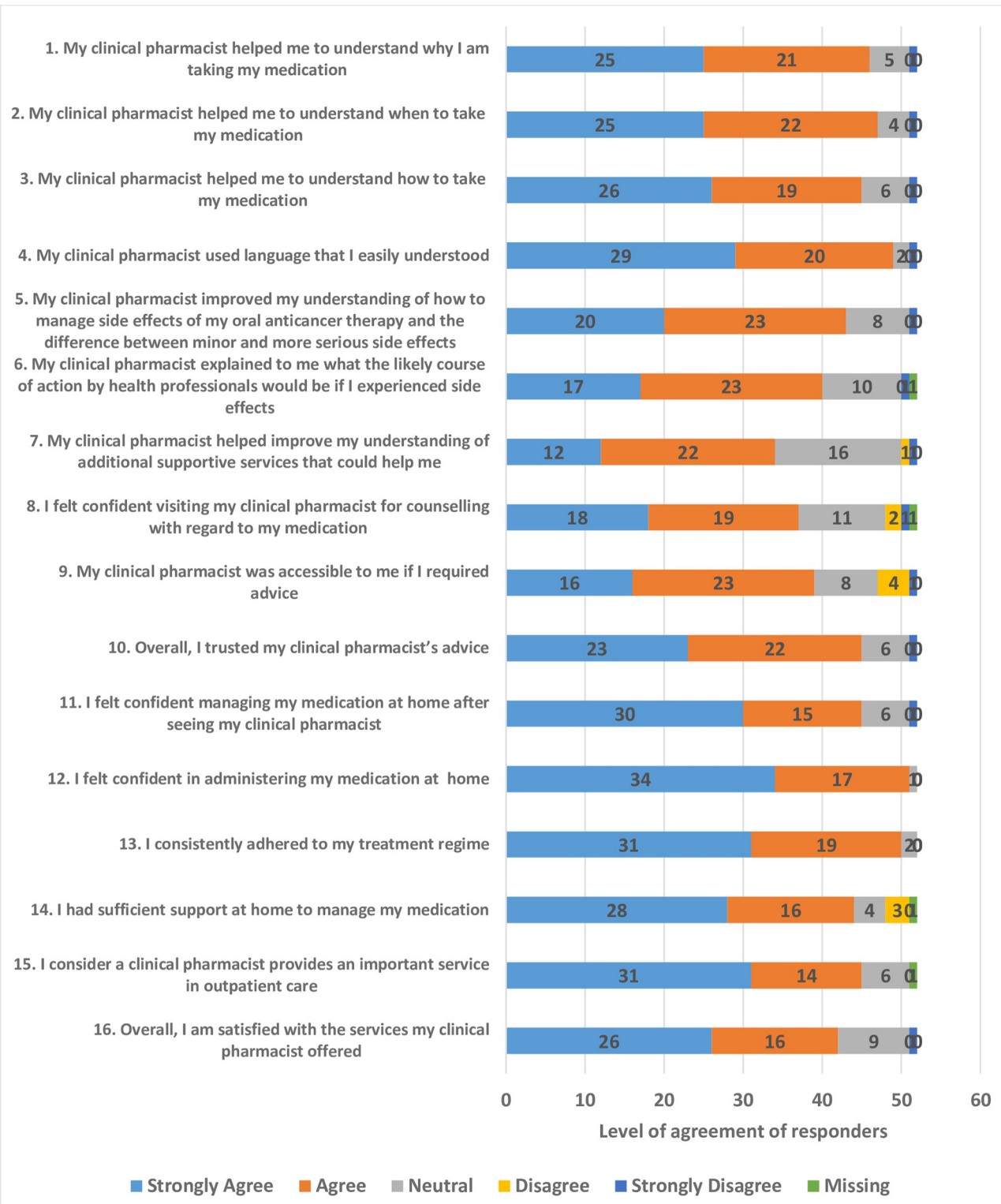

**Fig 1. Likert scale results of patient satisfaction with cancer services pharmacist consultation.**

**Table 2. Showing the numbers who agree with each statement amongst those who did or did not have a pharmacist follow-up.** P-values were obtained from the Chi-square statistic, unless otherwise marked.

| Statement | Pharmacist follow-up | | p value |
| --- | --- | --- | --- |
| | Yes | No | |
| My clinical pharmacist explained to me what the likely course of action by health professionals would be if I experienced side effects | 29/33 (87.9%) | 11/18 (61.1%) | 0.0370* |
| My clinical pharmacist helped improve my understanding of additional supportive services that could help me | 26/34 (76.5%) | 8/18 (44.4%) | 0.0209 |
| I felt confident visiting my clinical pharmacist for counselling with regard to my medication | 26/34 (76.5%) | 11/17 (64.7%) | 0.5075* |
| My clinical pharmacist was accessible to me if I required advice | 27/34 (79.4%) | 12/18 (66.7%) | 0.3338* |

* P-value obtained from Fisher's Exact test

## Discussion

To the best of our knowledge, this is the first Western Australian retrospective study assessing patient perceptions of a recently established dedicated CSP in an outpatient cancer clinic. This study found a positive trend in patients' perceptions, experiences, and overall satisfaction with the dedicated CSP at the Cancer Centre. This has also been reported in other studies [12, 13, 16, 17, 26]. Although positive trends occurred for patients perceptions from their experiences following counselling by a dedicated CSP at the Cancer Centre, it was after a follow-up that provided significant improvements. It would seem the first interview might be less effective when occurring amongst the milieu of a new diagnosis, or news of progression or recurrence of the patient's cancer, on top of all the information provided regarding their new OAA. However, when the first interview was followed up with a second CSP interaction, it was far more effective. This may also have been because patients have experienced some issues that they have an opportunity to resolve.

The questionnaire data suggests a causal relationship between the three interactions analysed (follow-up consultations, medication reconciliation and pharmacist intervention) and improved patient health literacy, improved understanding of how and when to appropriately take their medications and improved confidence in at home administration. Of these interactions, follow up consultations had the highest influence on patient perceptions of the CSP and was the most beneficial towards patient outcomes. Specifically, the interaction significantly improved patient understanding of when to take their medication (e.g., morning and/or night, cycle length, treatment length), improved their understanding of additional supportive services available and improved their confidence in at home administration. Furthermore, it was also found that these patients had significantly higher satisfaction with the CSP's services and increased trust in their CSP's advice. These findings are in line with the current literature that suggests pharmacist-led education sessions increases confidence and satisfaction with their OAA treatment [13, 17, 26, 27].

CSP's counselling sessions with patients undergoing OAA therapy have been shown to increase adherence, lower incidences of adverse effects, reduce treatment errors, improve therapeutic outcomes and enhance patient safety [1, 28–30]. An increase in patient satisfaction has been shown to be a vital contributing factor in determining whether a patient would seek advice from their CSP and maintain the patient-pharmacist relationship [20]. Moreover studies have shown pharmacist involvement can increase adherence and reduce occurrence of adverse effects [31]. These findings are in line with the current literature that suggests pharmacist-led education sessions increases confidence and satisfaction with their OAA treatment

[17, 26]. Positive patient experiences and satisfaction with their clinical pharmacist have been linked to increased adherence, improved patient-pharmacist relationships, and optimised therapeutic outcomes [1, 17, 28, 29]. In a recent study conducted in the United States, a closed-loop oral chemotherapy management program was implemented which employed a full-time equivalent dedicated speciality pharmacist [17]. The role of the pharmacist included educating patients about their oral chemotherapy, providing clinical patient management and improving patient adherence [17]. Researchers reported that in addition to the innovative model resulting in improved patient knowledge regarding their oral chemotherapy, improved adherence rates and greater patient satisfaction, it was also cost saving to the health system [17]. It is noted that in a recent systematic review by Maleki et al. [32] researchers reported that few studies have objectively assessed outpatient pharmacy cancer services. Given the recent emergence of a dedicated CSP within outpatient cancer clinics, particularly in Australia, it is important to raise awareness on their role and beneficial influence on patients undergoing OAA therapy.

Patients administered OAAs are at an increased risk of drug-drug interactions, non-adherence and adverse effects. Sufficient time spent with the CSP allows for appropriate counselling and patient concerns and/or questions to be addressed [13, 17, 20, 25, 30]. Further, a CSP can significantly improve patient medication adherence and laboratory monitoring [33]. Our findings advocate for the involvement of a CSP within OAA outpatient cancer clinics to improve patient outcomes, which is supported by findings from a previous study [14, 16, 17, 25]. To further strengthen these findings, the addition of a question regarding the number of follow-up interactions between the CSP and the patient would have been beneficial in gaining a deeper understanding of the association between the CSP and improved patient outcomes.

Questionnaire data also revealed a relationship with older age and positive responses. Of the three age categories with respondents (30–49, 50–69 and 70+ years), those aged between 50–69 years had the highest rate of 'yes' responses, followed by those aged 70+. This age bracket had significantly higher levels of trust in the CSP's advice, were more satisfied with and confident in the CSP counselling services. Existing literature comparing patient satisfaction with health care services are positively associated with older age [34–36]. These findings are in line with a study by Jaipaul et al. [34], where patient satisfaction with their health care providers increased linearly with age until 65–80 years and then declined [34]. The reasons for this remain unclear, however it could be due to health status, in that better health is positively associated with satisfaction or the differences in confidence, satisfaction and trust may be reflective of age-related expectations of their health care services. It is important to note that while these findings are still significant and relevant, the 50–69 year age bracket had the highest number of patients, accounting for 50.0% of the sample. This may be of value to CSP's and how they can tailor their counselling services to better suit this demographic. Further research into this correlation is necessary.

## Limitations

The findings of this study are limited by the localised patient sample, as patients were sourced from only one hospital. While the utilisation of the Likert scale questionnaire was effective, it did not account for the difference in comprehension and literacy levels within the participants. Additionally, the questions framed the role of the CSP positively, which may have led to acquiescence bias within the data. Finally, the recent rise in COVID-19 cases in WA had many impacts on our study. Firstly, communication with the hospital and the hospital supervisors was impeded and mostly conducted via email. Secondly, the pandemic lead to delays in the Australia Post services which meant questionnaires arrived at the patients' addresses later than

anticipated. This is assumed to have impacted the response rate and contributed to the smaller sample size. Further studies may benefit from utilising an electronic response system and an online questionnaire and reduce double handling of paper questionnaires. An electronic system may also aid in the data collection process.

## Conclusion

Overall, patients reported positive perceptions, experiences, and overall satisfaction with the CSP as part of their cancer support team during OAA therapy. Importantly, patients with more than one interaction with the CSP had significant improvements in their health literacy, understanding of when to take their medication and increased confidence in self-administration of their OAAs. The 50–69 age group may require improved tailored interactions. These findings should contribute to optimal therapeutic outcomes for outpatients.

## Author Contributions

**Conceptualization:** Bruce Sunderland, Richard Parsons, Tandy-Sue Copeland, Siobhan Corscadden, Selina Tong, Petra Czarniak.

**Data curation:** Lorna McNabb, Eva Metrot, Micaela Ferrington.

**Formal analysis:** Lorna McNabb, Eva Metrot, Micaela Ferrington, Bruce Sunderland, Richard Parsons, Petra Czarniak.

**Investigation:** Lorna McNabb, Eva Metrot, Micaela Ferrington, Bruce Sunderland, Tandy-Sue Copeland, Siobhan Corscadden, Selina Tong, Petra Czarniak.

**Methodology:** Lorna McNabb, Eva Metrot, Micaela Ferrington, Bruce Sunderland, Richard Parsons, Tandy-Sue Copeland, Siobhan Corscadden, Selina Tong, Petra Czarniak.

**Project administration:** Lorna McNabb, Eva Metrot, Micaela Ferrington, Tandy-Sue Copeland, Siobhan Corscadden, Selina Tong.

**Resources:** Bruce Sunderland, Tandy-Sue Copeland, Selina Tong, Petra Czarniak.

**Supervision:** Bruce Sunderland, Richard Parsons, Tandy-Sue Copeland, Siobhan Corscadden, Selina Tong, Petra Czarniak.

**Validation:** Bruce Sunderland, Petra Czarniak.

**Writing – original draft:** Lorna McNabb, Eva Metrot, Micaela Ferrington.

**Writing – review & editing:** Bruce Sunderland, Richard Parsons, Tandy-Sue Copeland, Siobhan Corscadden, Selina Tong, Petra Czarniak.

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
