## [Decision Letter · Decision Letter 0]

5 Feb 2024

PONE-D-24-00673Assessment of patient perceptions of counselling on oral antineoplastic agents by a dedicated Cancer Services Pharmacist in an outpatient cancer clinicPLOS ONE

Dear Dr. Czarniak,

Thank you for submitting your manuscript to PLOS ONE. After careful consideration, we feel that it has merit but does not fully meet PLOS ONE’s publication criteria as it currently stands. Therefore, we invite you to submit a revised version of the manuscript that addresses the points raised during the review process.

Please submit your revised manuscript by Mar 21 2024 11:59PM. If you will need more time than this to complete your revisions, please reply to this message or contact the journal office at plosone@plos.org. Please include the following items when submitting your revised manuscript:A rebuttal letter that responds to each point raised by the academic editor and reviewer(s). You should upload this letter as a separate file labeled 'Response to Reviewers'.A marked-up copy of your manuscript that highlights changes made to the original version. You should upload this as a separate file labeled 'Revised Manuscript with Track Changes'.An unmarked version of your revised paper without tracked changes. You should upload this as a separate file labeled 'Manuscript'.If applicable, we recommend that you deposit your laboratory protocols in protocols.io to enhance the reproducibility of your results. Protocols.io assigns your protocol its own identifier (DOI) so that it can be cited independently in the future. For instructions see: https://journals.plos.org/plosone/s/submission-guidelines#loc-laboratory-protocols. Additionally, PLOS ONE offers an option for publishing peer-reviewed Lab Protocol articles, which describe protocols hosted on protocols.io. Read more information on sharing protocols at https://plos.org/protocols?utm_medium=editorial-email&utm_source=authorletters&utm_campaign=protocols.

We look forward to receiving your revised manuscript.

Kind regards,

Yaser Mohammed Al-Worafi

Academic Editor

PLOS ONE

Journal Requirements:

Reviewers' comments:

Reviewer's Responses to Questions

**Comments to the Author**

1. Is the manuscript technically sound, and do the data support the conclusions?

Reviewer #1: Yes

Reviewer #2: Yes

2. Has the statistical analysis been performed appropriately and rigorously? 

Reviewer #1: Yes

Reviewer #2: Yes

3. Have the authors made all data underlying the findings in their manuscript fully available?

Reviewer #1: Yes

Reviewer #2: Yes

4. Is the manuscript presented in an intelligible fashion and written in standard English?

Reviewer #1: Yes

Reviewer #2: Yes

5. Review Comments to the Author

Reviewer #1: 1. This is an interesting study regarding patient’s perceptions about the CSP. Authors have mentioned that this is the first study in Western Australia. Are there any other studies from other parts of Australia? If so, please kindly reference them. Additionally adding systemic reviews, independent reviews and met analysis will add to comprehensive of your manuscript. Please kindly refer to one such review below. It will be worth to compare and contrast your results from the studies mentioned therein.(A systematic review of the impact of outpatient clinical pharmacy services on medication-related outcomes in patients receiving anticancer therapies)

2. Was permission sought from the authors to use their instrument?

3.Discussion: It supports the results with comparison from some countries only. Where there similar studies done in other parts of Australia as the author’s mention that this is first study in Western Australia.?

4. References: More references need to added

Reviewer #2: Thank you for giving me a chance to review this manuscript I have some comments that can improve the quality before acceptance

1- the design of the study is retrospective cross-sectional study why you mentioned cohort? there is no follow up period

2- some sentences in introduction have no reference.

3- there are several very long sentences that confused the reader please be more concise

6. PLOS authors have the option to publish the peer review history of their article (what does this mean?). If published, this will include your full peer review and any attached files.

Reviewer #1: **Yes: **Khizra Sultana

Reviewer #2: **Yes: **abdullah Salah Alanazi

---

## [Author Response · Author response to Decision Letter 0]

12 Mar 2024

Reviewer 1

1. This is an interesting study regarding patient’s perceptions about the CSP. Authors have mentioned that this is the first study in Western Australia. Are there any other studies from other parts of Australia? If so, please kindly reference them. Additionally adding systemic reviews, independent reviews and met analysis will add to comprehensive of your manuscript. Please kindly refer to one such review below. It will be worth to compare and contrast your results from the studies mentioned therein.(A systematic review of the impact of outpatient clinical pharmacy services on medication-related outcomes in patients receiving anticancer therapies)

Reply:

We have clarified that, although there have been a few Australian based studies, none of these studies investigated the impact of a dedicated cancer services pharmacist (CSP) consultations regarding oral anticancer medications. At the study hospital, a dedicated full-time CSP was employed on 1st January 2021 (as mentioned in the introduction). A previous study conducted in Australia in 2011 by Walter et al. (2016), demonstrated the benefit of a specialist cancer pharmacist in a lung cancer clinic. The pharmacist was employed for only six months. Further, two recent studies in Western Australia evaluated the impact of specialist pharmacist consultations in a comprehensive Cancer Centre across a range of cancer types. One of the studies, which also involved an online staff survey to evaluate perceptions of health service staff about consultations provided by CSPs, reported that 97% of staff strongly agreed/agreed that CSP consultations improved patients’ understanding of and confidence in managing their medications. Despite improved patient outcomes reported in both studies, neither study involved a dedicated CSP. The role of a dedicated CSP is a novel role in Australia and is presently poorly researched and acknowledged.

We have made reference in our introduction and discussion to a number of international studies. We have also compared and contrasted our results with other studies in the discussion. 

We appreciate the comment by the reviewer about the systematic review by Maleki et al. (Maleki S, Alexander M, Fua T, Liu C, Rischin D, Lingaratnam S. A systematic review of the impact of outpatient clinical pharmacy services on medication-related outcomes in patients receiving anticancer therapies. Journal of Oncology Pharmacy Practice. 2019;25(1):130-139). This systematic review was about the impact of outpatient clinical pharmacy services regarding patients treated with parenteral and oral cancer treatments. Although we have made reference to the systematic review by Maleki et al. in our discussion, as our study related specifically to oral antineoplastic agents, it was not possible to directly compare our findings with the systematic review by Maleki et al.

2. Was permission sought from the authors to use their instrument?

Reply:

The questionnaire was initially developed and utilised by authors (PC and BS) involved with the previous study by Dennis et al. In the previous study, a dedicated cancer services pharmacist was not employed at the hospital. The current study was conducted, using the previously developed questionnaire, to investigate the impact of a dedicated cancer services pharmacist following their employment at the hospital in January 2021. The supervisory team of the current study, PC and BS, ensured the questionnaire (which they were involved in developing and validating), was the same instrument used in the current study so that results could be compared.

3. Discussion: It supports the results with comparison from some countries only. Where there similar studies done in other parts of Australia as the author’s mention that this is first study in Western Australia.?

Reply:

Although global research into the impact of a CSP on outpatient OAA treatment is available and evolving, there are limited Australian based studies. A previous study conducted by the corresponding author at the same hospital, did not involve a dedicated cancer services pharmacist. Therefore, a gap exists in current literature on the benefits of a dedicated CSP in outpatient cancer clinics within the Australian health system. This is a novel role in Australia and is presently poorly researched and acknowledged.

4. References: More references need to added

Reply:

We have added a further eleven references to our manuscript. These are listed below as it did not appear to be possible to show the references as tracked changes in the manuscript:

1. Yeo HY, Liew ACi, Chan SJ, Anwar M, Han CH-W, Marra CA. Understanding Patient Preferences Regarding the Important Determinants of Breast Cancer Treatment: A Narrative Scoping Review. Patient preference and adherence. 2023:2679-2706. 

2. Dürr P, Schlichtig K, Kelz C, et al. The randomized AMBORA trial: Impact of pharmacological/pharmaceutical care on medication safety and patient-reported outcomes during treatment with new oral anticancer agents. Journal of Clinical Oncology. 2021;39(18):1983-1994. 

3. Holle LM, Puri S, Clement JM. Physician–pharmacist collaboration for oral chemotherapy monitoring: insights from an academic genitourinary oncology practice. Journal of Oncology Pharmacy Practice. 2016;22(3):511-516. 

4. Patel JM, Holle LM, Clement JM, Bunz T, Niemann C, Chamberlin KW. Impact of a pharmacist-led oral chemotherapy-monitoring program in patients with metastatic castrate-resistant prostate cancer. Journal of oncology pharmacy practice. 2016;22(6):777-783. 

5. Battis B, Clifford L, Huq M, Pejoro E, Mambourg S. The impacts of a pharmacist-managed outpatient clinic and chemotherapy-directed electronic order sets for monitoring oral chemotherapy. Journal of Oncology Pharmacy Practice. 2017;23(8):582-590. 

6. Suzuki H, Suzuki S, Kamata H, et al. Impact of pharmacy collaborating services in an outpatient clinic on improving adverse drug reactions in outpatient cancer chemotherapy. Journal of Oncology Pharmacy Practice. 2019;25(7):1558-1563. 

7. Kimura M, Go M, Iwai M, Usami E, Teramachi H, Yoshimura T. Evaluation of the role and usefulness of a pharmacist outpatient service for patients undergoing monotherapy with oral anti-cancer agents. Journal of Oncology Pharmacy Practice. 2017;23(6):413-421. 

8. Walter C, Mellor JD, Rice C, et al. Impact of a specialist clinical cancer pharmacist at a multidisciplinary lung cancer clinic. Asia‐Pacific Journal of Clinical Oncology. 2016;12(3):e367-e374. 

9. Sargent W, Whalley A. Implementation and outcomes of a pharmacist-led oral chemotherapy clinic at VA Maine Healthcare System. Journal of Oncology Pharmacy Practice. 2022;28(8):1704-1708.

10. Maleki S, Alexander M, Fua T, Liu C, Rischin D, Lingaratnam S. A systematic review of the impact of outpatient clinical pharmacy services on medication-related outcomes in patients receiving anticancer therapies. Journal of Oncology Pharmacy Practice. 2019;25(1):130-139. 

11. Megeed A, Magas H, Accursi M, Burant CJ, Hansen E. The impact of a pharmacist-led oral anticancer clinic on medication adherence and laboratory monitoring. Journal of Oncology Pharmacy Practice. 2023;29(8):1921-1927. 

Reviewer 2

1. The design of the study is retrospective cross-sectional study why you mentioned cohort? There is no follow up period

Reply:

We have removed the word ‘cohort’ from the manuscript.

2. Some sentences in introduction have no reference.

Reply:

We have added further references to sentences in the introduction.

3. There are several very long sentences that confused the reader please be more concise

Reply:

We have been careful to change very long sentences to shorter sentences to make them more concise. However, some of the official organisations or guidelines (eg the Australian Commission on Safety and Quality in Health Care, Clinical Oncology Society of Australia and the Society of Hospital Pharmacists in Australia) have long names. For clarity, it is necessary to include the complete name although this contributes to the length of the sentence.

---

## [Editor Report · Decision Letter 1]

6 May 2024

Assessment of patient perceptions of counselling on oral antineoplastic agents by a dedicated Cancer Services Pharmacist in an outpatient cancer clinic

PONE-D-24-00673R1

Dear Dr. Czarniak, 

We’re pleased to inform you that your manuscript has been judged scientifically suitable for publication and will be formally accepted for publication once it meets all outstanding technical requirements.

Kind regards,

Yaser Mohammed Al-Worafi

Academic Editor

PLOS ONE
---

## [Editor Report · Acceptance letter]

13 May 2024

PONE-D-24-00673R1 

PLOS ONE

Dear Dr. Czarniak, 

I'm pleased to inform you that your manuscript has been deemed suitable for publication in PLOS ONE. Congratulations! Your manuscript is now being handed over to our production team.

Kind regards, 

on behalf of

Professor Yaser Mohammed Al-Worafi 

Academic Editor

PLOS ONE